# Developing Students’ Emotional Competencies in English Language Classes: Reciprocal Benefits and Practical Implications

**DOI:** 10.3390/ijerph19116469

**Published:** 2022-05-26

**Authors:** Philippe Gay, Slavka Pogranova, Laetitia Mauroux, Estelle Trisconi, Emily Rankin, Rebecca Shankland

**Affiliations:** 1UER EN, HEP Vaud, 1014 Lausanne, Switzerland; laetitia.mauroux@hepl.ch (L.M.); estelle.trisconi@hepl.ch (E.T.); 2GRAFE, UNIGE, 1211 Geneva, Switzerland; slavka.pogranova@unige.ch; 3International Work Placement Coordinator, IAE, 38400 Grenoble, France; emily.rankin@univ-grenoble-alpes.fr; 4Laboratoire DIPHE, Lumière University Lyon 2, 69000 Lyon, France; rebecca.shankland@univ-lyon2.fr

**Keywords:** CLIL, socio-emotional competencies, academic performance, foreign languages

## Abstract

Learning a foreign language involves a wide range of cognitive, social and affective skills. The present article gives ideas to develop socio-emotional competencies in English courses: the capacity to identify the emotion, to understand the causes and consequences, to express their emotions and to do so in a socially acceptable manner, to manage stress and to use their emotions to increase the effectiveness of thinking, decision making and actions. Content and language integrated learning (CLIL) is a dual approach aiming to develop both language and academic subject knowledge. It may be gradually introduced, embedding it at three levels: into the classroom (routines, organization, pupils’ behavior), the school and the curriculum. Successful learning in CLIL remains based on (1) communication, (2) ways of engaging in the learning process and (3) the use of meaning-making strategies. We propose a pedagogical sequence (several courses) to learn a second language based on the social and emotional learning approach, and the English coursebook MORE! 7e for primary school pupils (aged 10–11). We combine the specific language learning of the unit—talking about ourselves, people and their feelings—with the development of pupils’ basic emotional competencies, and discuss advantages and disadvantages to consider in order to successfully implement such lessons.

## 1. Introduction

Emotions are omnipresent in everyday life and throughout one’s entire life. Generally speaking, emotions are characterized by a strong intensity, a short duration and occurrence due to a clearly identifiable stimulus; in contrast, moods are of a weaker intensity and longer duration, and sometimes appear without a specific trigger. Emotions generate different reactions, notably physiological and behavioral (see e.g., [1,2] for more details regarding the components of emotions and other affective phenomena). Therefore, the way in which individuals gradually learn to manage pleasant and painful emotions has an impact on their physical and mental health. For example, cardiovascular and psychological health, as well as professional success, academic performance and harmonious social relationships, can be promoted by good emotional competencies, as characterized by the ability to identify, understand, express, use and regulate one’s own emotions and those of others (for a review, see [3,4]).

Learning English as a second foreign language involves a wide range of cognitive, social and emotional skills. In the classroom, students experience emotions in various classroom situations, such as frustration, worry, disappointment and boredom, but also hope, enthusiasm and pride. Dealing with these emotions is crucial to ensuring the quality of academic learning, as well as the quality of life [3]. Students’ emotions during the lesson depend, in part, on the classroom climate, relationships with classmates, difficulty of tasks, success in learning, teacher’s supportive attitude and attitude towards languages (e.g., [5]). For example, some students might feel uneasy or bored as second or third language learners when they have to memorize vocabulary and find this difficult. Making students aware of vocabulary-learning strategies linked with positive emotional reactions (e.g., interest, pride) may help them to overcome difficulties and improve language learning. Indeed, emotions (e.g., emotions, feelings) are closely related to aspects of cognition and behaviors [6], and are therefore increasingly considered in teaching and learning processes [7,8]. Courses in the school curriculum that are directly aimed at developing these skills are, however, lacking in Switzerland’s curriculum [9], as well as in teachers’ practices in France [10].

Content and language integrated learning (CLIL), which includes emotions in the course content, appears as a promising solution in order to work on emotional skills without delaying the work on the classroom curricula. The addition of CLIL and social and emotional learning (SEL) enables students to learn a language as well as knowledge related to social and emotional competencies, and to apply these competencies directly in the classroom.

In the present article, in order to facilitate the comprehension of our pedagogical project presenting CLIL as an opportunity to balance curriculum and SEL, emotional intelligence and emotional competence will be considered as synonymous, and will be referred to as SEL. Based on unit 3 of the English coursebook MORE! 7e [11] for primary school students (aged 10–11) in the region of Geneva, we propose a pedagogical sequence that leads teachers and students through an educational project based on the SEL approach. We combine the specific language learning of the unit—talking about ourselves, people and their feelings—with the development of students’ basic emotional competencies.

Drawing on knowledge from CLIL-based literature and SEL, we seek to develop tools that may be used by teachers and that lead their students towards positive self-perceptions while learning English. The forms and the content of our material aim to drive students to the heart of complexity, and to lead them to be involved both cognitively and emotionally through specific exercises. Our sequence is linked to the current research on emotions in teaching and learning [7,12], and, more specifically, in foreign language learning [13,14]. The article ends with recommendations for teacher training courses [15].

## 2. Emotions in Teaching and Learning

In the school context, and, more particularly, in relation to learning, negative emotions have been extensively studied as factors that interfere with success (e.g., [16,17,18]). There are currently studies in the field of neuroscience that continue to improve the understanding of the mechanisms underlying the deleterious effects of negative emotions on cognitive functions (e.g., [19]). More recently, research is also increasingly considering the importance of positive emotions (such as joy, interest, pride) in the learning process [20,21].

As far as the school’s classroom teaching is concerned, regardless of any courses or subjects, teachers can carry out teaching work around three objectives: first, developing skills and knowledge; second, working on emotions, attitudes and values; and third, making the students conscious of the learning process and strategies. The emotions intervene in the process of helping them to achieve these goals, and the integrated learning is part of the teaching. In the learning of second or third languages, emotions may play a crucial role, as language learning requires daring to express oneself even if one does not know the right words, and possibly speaking or writing with errors, as well as managing the stress of difficulty in understanding what other are saying. They can help achieve the learning objectives in an English class, such as productive (speaking/writing) or receptive skills (listening/reading), acquiring the knowledge about the language (e.g., spelling, grammar) and the meaning-making strategies to understand, speak or become aware of the learning itself (learning seen as a process or a result).

## 3. Content and Language Integrated Learning

CLIL is a dual approach aiming to develop both language and academic subject knowledge. As the content and the language are involved, it means taking into consideration the language acquisition approaches and CLIL methodology. Based on the 4Cs Framework [22,23], four components are incorporated into a CLIL lesson: content, communication, cognition and culture. The content refers to the topics and the themes as defined in the curricula; the communication is the language that students need in order to communicate; the cognition refers to the thinking; and the culture refers to the interaction with the knowledge (social awareness). This framework has been widely used in past research. A CLIL methodology consists of an integrated approach where students learn to use the language, but also use language in order to learn [24]. More precisely, the language fulfils three functions in Coyle’s (2007) analytic framework: language for, of and through learning. The first refers to the metacognitive skills for effective learning: students apply them when memorizing and during cooperative work and debates. The second is needed to access the concepts in order to scaffold the content of the subject matter. The third appears during the interaction and combines deeper learning with higher thinking skills while students are cognitively engaged in tasks. Coyle states that the effectiveness of CLIL is through the “progression in knowledge, skills and understanding of the content, engagement in associated cognitive processing, interaction in the communicative context, the development of appropriate language knowledge and skills as well as experiencing a deepening intercultural awareness” (p. 550).

As we aim to present a pedagogical sequence, it is interesting to consider CLIL from the teacher’s perspective. The way that the teachers look at the content and the language can be familiar or new: the teachers present students with familiar content in the introductory phase, followed by specific vocabulary needed for the content in order to overcome language barriers ([22] quoted in [25], p. 52). Later, during the main stage, new content is introduced while the language remains accessible to the students. Finally, during the extension activities, the objectives focus once again on language, on a more complex level. According to Marongiu [26], one of the specific features of CLIL is that teachers “provide scaffolding strategies for the learners so as to help them process content and do something with it, such as use it to solve problems, to hold a position in discussions, to create a text, or to develop a project etc.“ (p. 83). For this aim, the author lists strategies helping with content processing and manipulation: simplifying, paraphrasing, explaining, adding language frames, word banks, diagrams, pictures, etc. The activities are carefully selected by the teacher or invented so that learners are “disengaged from linguistic difficulties” and involved in tasks requiring “effective communication” (p. 88). Successful learning in CLIL therefore remains based on communication, ways of engaging students in the learning process and in using meaning-making strategies [27].

CLIL lessons present their advantages and pitfalls [28,29], which should be carefully considered in order to implement such lessons successfully. Darvin et al. (2020) underline that CLIL has been seen as a kind of in-betweenness; for example, between language teaching, which is explicit or implicit, and between a focus on form or meaning, which has allowed for many interpretations. Following these authors, another point is that a balance of content and language is a rare practice, resulting in different roles taken by language and subject teachers. The balance is something we aim at in the sequence, considering English as a means of instructing and simultaneously as a learning objective. In his article on reasons for and against CLIL, Bruton (2013) highlights that students’ immediate needs do not necessary match with what they are taught in schools and that “[t]his gain in the sense of purpose in CLIL can only be verified by surveying students impartially in terms of what content, if any, they would prefer to study in a FL (Foreign Language) or even that they might find more beneficial… not their parents or the authorities“ (p. 591). Interestingly, any content of CLIL seems interesting with regard to developing social and emotional competencies. The results of a large-scale study in Spain [30] indicated that the CLIL approach in general (i.e., without specific themes) could positively influence the acquisition of emotional skills in secondary education. In comparison to non-CLIL classes, students who were taught CLIL were significantly more emotionally competent than their peers, leading the researcher to conclude that there is a connection between the CLIL approach in general and the development of emotional competencies, such as emotional awareness, emotion regulation and social awareness [30]. This could be related to the fact that students who are taught CLIL show better communication skills, such as strategies related to expression and understanding. At primary school level, we need to consider whether and to what extent the students may express their emotions and talk about others. This includes their interest in the topic and their cognitive skills.

## 4. Swiss Context

The specificity of our approach lies in the fact that the primary focus is on the content taken from the model of emotional competencies [4] and the curriculum [9]. The latter implies that, within the theme of health and well-being, students become conscious of themselves on the physical, social and emotional level and make decisions accordingly. The curriculum stipulates that emotions are brought up according to students’ needs. Different behaviors and their consequences are discovered and, finally, different emotions are identified and differentiated, and students learn strategies to regulate them. Usually, emotions do not represent a teaching subject as such (e.g., in Switzerland), but their components are part of school subjects, including English. In the French-speaking part of Switzerland, English follows German and is taught to students aged 10–12 as a second foreign language during the last two years in primary school. We were inspired by the teaching material used in English as a foreign language class (MORE! 7e, 2013 [11]), and its specific exercises combined with the above mentioned cross-curricular and theoretical dimensions. The sequence is designed for the unit 3, My friends, situated in the first half year. As students are at early stages of learning English, the language scaffolding plays an important role. The implementation of the sequence can be performed by any school teachers paying attention to the ways that the content input activities are made accessible to students. It can also be used by English language teachers who add content to their language-related exercises.

## 5. Classroom Aims: Emotional Competencies in English Classes

The implementation of social and emotional skills depends on a wide range of skills. For example, managing one’s own and others’ emotions involves identifying, recognizing, and understanding emotions. Regulating emotions may then enable being able to express one’s own emotions or listen to others’ emotions, and to draw essential information from them to make appropriate decisions. A useful taxonomy for both teachers and students has been developed to present this range of skills.

The model of emotional competencies developed by Mikolajczak et al. [4,31] distinguishes five basic emotional competencies (see Table 1), each one being related to an intrapersonal level (self) and to an interpersonal level (others): ability (1) to identify emotions, (2) to understand the causes and consequences of emotions, (3) to express emotions and facilitate other’s expression of emotion, and to do so in a socially acceptable manner, (4) to regulate emotions (when inappropriate according to the context) and (5) to use emotions to increase the effectiveness of thinking, decision making and actions.

In the context of this five-facet model, in order to improve emotional competencies, one must first be able to identify emotions, both in oneself and in others. In other words, it is helpful to be able to distinguish emotions and name them in order to understand them better and act in an adapted way to the situation at hand. For example, it is more complicated to know how to act when faced with a friend who is experiencing difficulties if we cannot identify whether they are feeling anger, sadness, fear, tiredness or shame. In order to develop these skills of decoding oneself and the other’s gestures or facial expressions, past research has underlined the benefits of developing a better “granularity” of emotions (e.g., irritated, annoyed, angry, exasperated, enraged…). It is also useful to combine this work on one’s dictionary of emotions with the perception of bodily sensations: what happens at the physiological level in my body when I feel, for example, angry, sad, ashamed, surprised, frightened, disgusted or joyful? How does my heart beat? Which parts of my body feel warmer? How does my breathing change? In addition, in the face (e.g., eyes, eyebrows, mouth) or voice, what can I notice in others that would help me to differentiate these emotions?

After having identified an emotion, it is useful to understand its causes and consequences. This is the second pillar of emotional competencies to be shaped: finding the origin of an emotion and the reason why it is present. Indeed, emotions usually play an adaptive role in the sense that they are a reaction to a situation considered, such as needing attention. Sometimes, the emotion triggered by a situation (e.g., a critique that makes one sad) is then increased because of a deeper cause (e.g., several weeks of harassment). In this context, research has underlined the benefits of various educational programs (e.g., [12,32,33]) that focus on a better understanding of how emotions work, and include training that allows children to learn to link situations and emotions by confronting their point of view with those of others.

Emotional competencies also require adapting expressing one’s own emotions and listening to those of others or helping others express their emotions. Verbalizing feelings appropriately (e.g., using language that is not demeaning to others, moderating the intensity of speech, finding the right moment) is often difficult, and different communication methods can help school age children (for example, it is possible to follow the four steps of nonviolent communication [34]: (1) calmly describe the situation in a factual, non-judgmental way: say, e.g., “there are several classmates still waiting for a copy of the document” rather than “you always want to help yourself first”; (2) express one’s own feelings by using the pronoun “I” and avoiding “you”: say, e.g., “I feel stressed and I can’t concentrate when there is noise in class” rather than “you disturb everyone”; (3) explain the needs that are not being met: say, e.g., “I feel irritated because I had planned to have time to continue my reading”; (4) propose a concrete solution (or ask what the other person suggests to solve the problem): say, e.g., “next time, it would be nice to let me finish before coming to ask me a question” rather than “please make an effort from now on”). The development of these five competencies—to identify, to understand causes and consequences, to express, to regulate and to use—could be proposed in SLA (second language acquisition) lessons. In this article, we have taken up the existing pedagogical resources (MORE! 7e, 2013, [11]) to make explicit the competencies worked in the proposed activities; we also propose extensions and additional activities specifically related to each emotional competence.

As a taxonomy of the different social and emotional competencies, we think that the model presented in Table 1 may be useful for various audiences, such as teachers, researchers and policy makers.

## 6. Pedagogical Sequence

This section presents objectives and ideas to implement a pedagogical sequence in English as a second language based on a socio-emotional learning (SEL) approach for primary school pupils (aged 10–11).

Research has shown that the identification of mixed emotions (e.g., I am both anxious and happy to give this talk about my favorite sport) progresses between the ages of 7 and 10 (e.g., [35]). However, other research has highlighted that 15-year-olds still have a fair amount of difficulty recognizing fear and sadness (e.g., [36]). It could therefore be particularly interesting to enrich the emotional lexicon, even more so from the age of 10–11 years and in a language context. For example, students have the opportunity to learn vocabulary from the new language while learning to better identify their own emotions. This work on the granularity of emotions, for example, in terms of their valence (from unpleasant to pleasant) and their intensity (weak to strong), can be carried out via a grid (mood meter) that allows the teacher and the children to identify the mood they are feeling [37,38]. These elements may be considered as basic pillars to assist in the regulation of emotions, which can be difficult, especially during adolescence (e.g., [39]).

### 6.1. Ability to Identify and Understand One’s Own Emotion

Based on the five-facet model presented above [4,31], examples of objectives for the first two competencies are suggested below.


**Objective related to the intrapersonal SEL of emotion identification:**
The students are able to identify and describe the sensations in the body and the facial muscles related to specific emotions. This can be related to parts of bodily sensations (e.g., heart, breath, skin).



**Objective related to the intrapersonal SEL related to understanding the emotion:**
The students are able to make a guess about an emotion by describing orally, by picture, by miming or by drawing the situation that caused the feeling and the consequences of this emotion.


More concretely, the language exercise that follows is situated at the beginning of unit 3 (MORE! 7e, 2013, [11]) and constitutes a good starting point to address these skills. The emotion—to be scared—is introduced in a situation where two children meet a new neighbor and her dogs. One of the children is scared of dogs. In the communicative context, students identify the emotion and understand its meaning. It is not important to understand every word of the text. Students listen and read (ex.1), try to understand the gist and complete a gap-fill exercise (ex.2) with appropriate names. The comprehension is checked and students’ answers are justified. The sentence “I’m scared of dogs” is written on the board and memorized, and its pronunciation is drilled.

The expression of one’s own emotions is a topic in English class found in unit 3 of the English coursebook MORE! 7e [11]. In order to identify what students feel, the teacher introduces flashcards with emotions—hungry, bored, angry, cold, sad, scared, hot (ex.15, p. 29). Students repeat after the teacher, memorize them and accompany the language learning with gestures and facial expressions. A memory game is played where written words and pictures are assembled and the sentences are pronounced by students aloud (ex., I’m cold). Afterwards, students read scrambled sentences (ex.14, p. 29), and put them in the right order (individually or in pair work). The pictures help them to understand the situation, i.e., where such dialogues could take place. The recording is played to check the right order of sentences: How are you today? I’m not very happy. Oh, dear. Why? It’s Monday morning! Exercise 15 follows. The teacher, with the help of English-speaking students, make a few examples of similar dialogues, using another emotion from exercise 15. The students are put in groups, and they choose an emotion and act out dialogues. The language in both exercises allows them to produce several dialogues with different combinations. Students’ speaking skills are developed in an amusing and constructive way. Eventually, they create new dialogues, insert other words (emotions not yet identified) and draw pictures. The scenes and the pictures are presented to the class, and the students then vote for the most funny, original or interesting scene.

### 6.2. Ability to Identify and Understand the Emotions of Others

Based on the five-facet model presented above [4,31], examples of objectives for the first two competencies could be as follows.


**Objective related to the interpersonal SEL of emotion identification:**
The students are able to identify the emotions of others based on pictures, video and audio (see e.g., https://greatergood.berkeley.edu/quizzes/ei_quiz/take_quiz, accessed on 5 April 2022; https://www.proprofs.com/quiz-school/story.php?title=facial-expression-recognition-test, accessed on 5 April 2022; http://www.unige.ch/cisa/properemo/minipons/demo.php, accessed on 5 April 2022);The students are able to distinguish a fake and a genuine smile (see e.g., https://www.surveymonkey.com/r/SmileRead, accessed on 5 April 2022).



**Objective related to the intrapersonal SEL related to understanding the emotion:**
The students are able to make a guess about an emotion by describing orally, by picture, by miming or by drawing the situation that caused the feeling and the consequences of this emotion in relation to themselves.


The teacher introduces animals (snake, mouse, spider) and students listen to recordings of other children describing which animals they are scared of (ex.9, p. 27). They listen and match the children’s fear with animals. This allows them to be exposed to the new language, presented in new dialogues and to check their comprehension. They have an oral model, where they hear the melody, the accents and the intonation. After a few encounters with the language, the questions and answers are the focus of the lesson: Are you scared of snakes? Yes, I am. No, I’m not. The objective of the speaking task (exercise 10) is to identify their emotion when they think of different animals. Students take the role of children on the CD and later answer the questions for themselves, which relates to intrapersonal identification. The complexity of the language increases as they juggle between the first and third person, emotions, the questions and the answers, which might be a challenge for primary school students. A class survey can be conducted as well until students memorize the content (who is scared of what) and develop their language skills. As a complement to this exercise, teachers could propose a granularity of the emotions of fear and give specific vocabulary for different intensities of this emotion (e.g., he/she is terrorized, panicked, frightened, worried, concerned…).

The language exercises are focused on the emotions of others. The vocabulary learnt so far is expanded and a new vocabulary input is carried out. Students match the pictures with written words, listen to the recording and repeat the sentences (ex.6, p. 26). The focus is now on memorizing new sentences using the third person: Owen is happy. Students realize that others can feel different emotions and that they can express them in a meaningful way. The knowledge is consolidated by identifying the emotion in the pictures, encircling the word in a snake and describing people in written form (Workbook, ex.4, p. 17). This is followed by an interaction between students. The questions are asked about how others feel (Is Maria tired?) and the examples of answers are given (No, Maria is cold). Several questions and answers are formulated by students, the complexity of the language is dealt with by comparing the oral and written forms and the teacher can add other pictures and emotions as a differentiated activity. The content and the language are worked on in a balanced way.

The expressions of emotions in the target language can then be enlarged in a plurilingual activity. Students are asked to work in groups and to create a poster where pictures of their choice are pasted and what people feel is described in different languages. The posters are then presented, read and discussed with the class. This can add an intercultural dimension to the learning. From the language point of view, the sentences in English, French and German are compared and observed, and notes are made on the similarities and differences (ex.8, p. 26). Another possibility is to link English with art. In the art lesson, students make a project in which people’s faces are drawn and special materials are used. Based on the artistic work, students express the desired emotions in English and discuss them in French. Creating artistic works that properly convey emotions can be difficult. Teachers should thus focus on discussions that can follow such work; for example, they can ask “what makes you say that this work is related to these emotions? How would you have done it yourself to express these emotions?”.

The language exercises become more complex and aim to include everyday life. Students listen to the description of Lily’s week (ex.17, p. 30). This description combines activities, days of the week and emotions. A new vocabulary input is carried out—students listen and repeat the days of the week, which are then compared to French (school language) and German (first foreign language learned in primary schools) (Workbook, exercise 5, p. 17). The exercise allows them to raise awareness of possible similarities (Monday—Montag) and develops their metacognitive skills. Students talk about Lily (ex.18, p. 30) and her emotions every day: Lily is cold. It’s raining. It’s Thursday. How does she feel? Students realize that emotions can change and be different according to the situations and circumstances, which is related to the competence of understanding emotion. They discover the everyday life of children living in other countries (ex.21, p. 31). They identify specific information (Is Fadry at school?), including how the children feel, linked to the identification of emotion. These cross-cultural texts are presented in the book in the form of a website that students could encounter in their everyday life. Finally, students read about Tim and describe his week (ex.22, p. 31). They name a day and the emotion, and explain why Tim is feeling that way. This allows them to link the emotions with the causes, and thus understand others’ emotion. They can also find a picture of a friend and write a few sentences about their feelings. They can interview a friend or write about a pop star’s activities and feelings. In this project, students show what they have learned in unit 3.

Although basic emotions appear to have universal facial expressions (see e.g., [1]), the meaning of specific gestures can be significantly different according to the country or culture. For example, in India, they tend to say yes by shaking their head as if they were saying no in a Western country. Giving students some examples helps them to become aware of cultural differences in expressing certain emotions and in the evaluation of a situation that can lead to very different emotions.

### 6.3. Abiliy to Express One’s Own Emotions in a Socially Acceptable Manner

Based on Mikolajczak et al.’s model [4,31], objectives for the expression of emotions could be as follows (asking students to express their emotions may cause them to say something other than their real emotions or conform to the majority’s responses. In some cases, showing certain emotions could lead to bullying. A crucial point may be to create a supportive classroom environment by setting clear rules (and enforcing them) in particular exercises. Non-violent communication and the expression of positive emotions of gratitude are easier to implement, and work towards the installation of harmonious social relationships and an optimal classroom climate (e.g., [33,34])):The students are able to express gratitude to someone who helped them or to write a heartfelt message of gratitude to a loved one (e.g., [40]);The students are able to create an appropriate response, using the principles of non-violent communication [34], for the person who broke their phone (see Figure 1).

Making emotions an issue for students is meaningful to them. The pedagogical sequences as presented in this article are an example of a didactic device that can be implemented for this purpose. After having presented the sequences, we will now underline the arguments on which their development is based. This work enables the students to develop their ability to better apprehend the concept of emotions through two dimensions of emotional competencies: identifying emotions and understanding the potential causes and consequences of emotions. In addition, the sequence is designed to deal with both intra and interpersonal skills. It is also important to underline that, in addition to the quality of relationships and adaptability to one’s environment, emotions are responsible for giving rise to dispositions to learn, even proving to be a key ingredient in the success of one’s learning. Emotions caused by mastering the school task situation are associated with better study strategies [41], and generally come with the need to expand knowledge. Emotions such as anxiety, anger, tension, despair and frustration appear when the discrepancy is too important between the competencies required by the task and the student’s current level of competence. These emotions are associated with strategies that aim to maintain a sufficient level of well-being and reduce the student’s engagement in learning strategies [42]. It is therefore essential to take this into account in order to encourage students’ engagement in learning [43]. Another way of saying this is that, if learners are anxious (“I take a long time to concentrate on something”) or show higher levels of hesitation (“I don’t know where to start”), these orientations may affect the course of thoughts, and adversely degrade information processing and decrease their performance [44].

The degree of mastery of a task represents one source of emotions for the learner, but there are numerous situations in schools that generate emotions that can increase or reduce learning engagement (see e.g., [45]). Furthermore, emotions caused by a school situation do not all originate from a school issue, just as the school experiences are likely to reverberate outside of the school setting alone. This can be explained by the multiplicity of psychological needs, such as needs for autonomy, social belonging and competence [5], that may be expressed in an individual simultaneously. Indeed, they are expressed in relation to what the student experiences and the way in which they perceives themself as an individual who learns, socializes, develops intimate friendships and physical capacities and adapts to their physical appearance, all at the same time. The educational material presented, which simulates everyday situations of surprise, joy or sadness, takes this complexity into account by relying on a global approach to emotions. It considers the individuals, including all of their basic psychological needs, without being limited to the targeted academic achievement.

Presently, more specific courses targeting the development of emotional skills in connection with cognitive engagement (e.g., perseverance, sustained attention) are useful to implement, as emotion and cognition are interrelated. Developing emotional competencies for the benefit of greater confidence in one’s chances of success is a useful objective to aim toward for the students. This can be carried out in the classroom by proposing self-regulation strategies; for example, by encouraging oneself, by remembering successes or by taking concrete action to reduce physiological tension (e.g., relaxing, walking; [44]).

The device presented is inspired by the CLIL methodology, which gives it a hybrid character. It not only builds on the CLIL principles, namely the development of language skills through the triangulation of cognition, culture and communication, but also introduces learners to positive emotion management. Working on emotions, through the elementary verbalization processes that it implies, combines particularly well with didactics, in which, the development of language skills predominates as learning objectives.

## 7. Conclusions

Several approaches, such as RULER (an acronym for recognizing, understanding, labeling, expressing and regulating emotions), exist to help teachers and educators develop their own emotional competencies and teach emotional competencies to children from kindergarten through high school [12]. Such evidence-based programs include training teachers to understand how emotions enhance learning, relationships and well-being. These trainings also incorporate specific tools, activities and lessons to develop teachers’ and students’ emotional competencies. The training of future teachers can more explicitly take up the five objectives related to Mikolajczak et al.’s model [4,31], and proposes, for each, competence and inter and intra-personal exercises. In order to fit more adequately with a CLIL approach, the training can also emphasize comparisons between different cultures (e.g., Ekman’s seven universal emotions; cultural differences in emotion expression). If teachers of foreign languages want to work on both curriculum and the development of such overarching skills, the CLIL approach appears to be a promising match.

## Figures and Tables

**Figure 1 ijerph-19-06469-f001:**
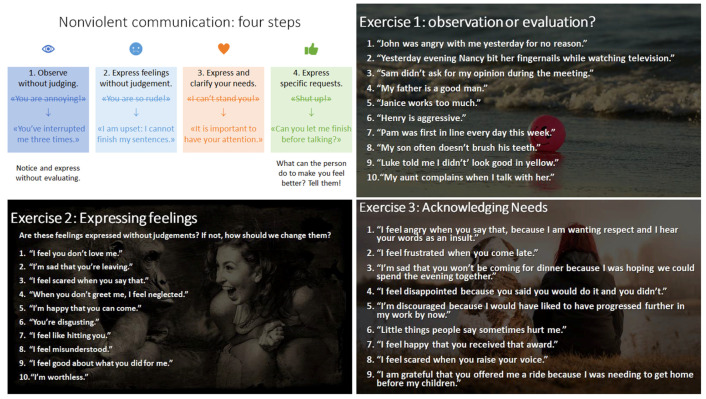
Examples proposed in a pedagogical session built for the Masters thesis of Marchetti and Reymond (adapted with permission from [35]). 2021, Marion Marchetti.

**Table 1 ijerph-19-06469-t001:** Five basic emotional competencies in their interpersonal and intrapersonal aspects (adapted and translated with permission from Mikolajczak, M.; Quoidbach, J.; Kotsou, I.; Nelis, D. Les compétences émotionnelles, New presentation © 2020, Dunod Editeur, Malakoff).

	Intrapersonal Side (Self)	Interpersonal Side (Other)
	Individuals with high emotional competence...	
Identification	… are able to identify their emotions.	… are able to identify the emotions of others.
Understanding	… understand the causes and consequences of their emotions.	… understand the causes and consequences of others’ emotions.
Expression	… are able to express their emotions, and to do so in a socially acceptable way.	… allow others to express their emotions.
Regulation	… are able to manage their stress and emotions (when these are inappropriate to the context).	… are able to manage stress and emotions of others.
Use	… use their emotions to increase their efficiency (in thinking, decision-making, actions).	… use the emotions of others to increase their efficiency (in thinking, decision-making, actions).

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
