# Peer review of "Developing Students’ Emotional Competencies in English Language Classes: Reciprocal Benefits and Practical Implications"

_ijerph, 2022, doi:10.3390/ijerph19116469_

Round 1

Reviewer 1 Report

I found the article very interesting and overall well presented, I think the conceptual tools used (eg the Mikolajczak et al. model) may be useful for various audiences such as teachers, researchers and policy makers .

I would like to suggest to make the rationale even sharper providing a more detailed breakdown ( maybe some examples?) of the contexts in which 2nd and 3rd language learners find themselves unease and explaining how crucial the emotional reaction may be to overcome difficulties and develop learnings rather than frustrations which risk to have a permanent impact on behaviors  (lines 42-59).

I would place  part of the sentences concerning "identifying emotions" before the Mikolajczak et al. model, in order to better introduce the reader to the usefulness of the model.

Author Response

Response to the first review: Thank you very much for the constructive feedback. Please find hereafter responses and changes corresponding to each comment.

  • We added some examples and descriptions to underline the role of emotions in 2nd-3rd language learning as follow:

For example, some students might feel uneasy or bored as second or third language learners when they have to memorize vocabulary and find this difficult. Making students aware of vocabulary learning strategies linked with positive emotional reactions (e.g., interest, pride) may help them overcome difficulties and improve language learning.

  • As suggested, we added this paragraph at the beginning of section 5 to better introduce the reader to the usefulness of the model:

The implementation of social and emotional skills depends on a wide range of skills. For example, managing one's own and others' emotions involves identifying, recognizing, and understanding emotions. Regulating emotions may then be being able to express one’s own emotions or listen to others' emotions, and draw essential information from them to make appropriate decisions. A useful taxonomy for both teachers and students has been developed to present this range of skills.

  • We also added that conclusion at the end of section 5:

As a taxonomy of the different social and emotional competencies, we think that the model presented in Table 1 may be useful for various audiences such as teachers, researchers and policy makers.

Reviewer 2 Report

  1. The title of the paper does not quite match the content of the paper.

As a second language English subject, not all course content in a semester course is suitable to penetrate the emotional and emotional education goals, and all are suitable for the development of students' emotional intelligence. It is suggested to change the title of the text to "Exploration of Teaching Lessons for Developing Students' Emotional Competence in Second Language Teaching——Take the "my friend" lesson as an example.

2. The content demonstrated in the article is not in line with the current teaching practice.

As far as the school's classroom teaching is concerned, regardless of any courses or subjects, teachers should carry out teaching work around three objectives in the design and arrangement of teaching goals in a class: first, knowledge and skill objectives; second, Emotional, Attitude, and Values objectives; third, process and method objectives. The emotion, attitude and value goals in the teaching process of teachers need to be penetrated in the process of helping students achieve knowledge and skill objectives. Therefore, the integrated learning of subject teaching tasks and content is not the characteristic or distinctive feature of English courses, but the teaching objective that should be achieved in the normal classroom teaching of any subject.

3.The design of the Pedagogical sequence and some contents do not conform to the starting point of the students' teaching.

From the perspective of the development characteristics of students' individual emotions, individuals have developed their emotions and emotional abilities in their different living environments, learning environments and learning in different disciplines since birth. Individual social emotions are generated on the basis of cognition, which is a subjective attitude experience in the process of learning knowledge, skills and daily life events. By the time they reach the age of 10-11, they should already have a certain emotional cognitive foundation and emotional abilities that can be transferred to second language learning, instead of the zero foundation in this article. Therefore, for truly meeting the development needs of 10-11-year-old students' emotional ability, Pedagogical sequence needs to be improved and adjusted according to specific contents in (1) identify emotions, (2) to understand the causes and consequences of emotions. Such as sad, scared and happy, etc. Ten-year-old students should generally reach a more competent level in recognition and attribution.

  1. In the introduction and literature review of the paper, some expressions are unprofessional, knowledge gaps are obvious, others are not quoted, and the information conveyed is unconvincing.

4.1“Emotions are characterized by a strong intensity, a short duration, an occurrence due to a clearly identifiable stimulus, in contrast with moods which are of weaker intensity, longer duration and sometimes appear without a specific trigger.” Please find the citation basis for this sentence.

4.2 “Therefore, the way in which individuals gradually learn to live and manage pleasant and painful emotions has an impact on their physical and mental health. ” Live is used here with ambiguous meaning.

4.3“Indeed, emotions (e.g., emotions, feelings) are closely related to aspects of cognition and behaviors [6], and therefore are increasingly considered in teaching and learning processes [7,8]. Courses in the school curriculum that are directly aimed at developing these skills are however lacking.” Please find a valuable citation basis for the last sentence. (both theoretical and practical, if possible)

4.4“In comparison to non-CLIL classes, students who were taught CLIL were significantly more emotionally competent than their peers, leading the researcher to conclude that there is a connection between the CLIL approach in general and the development of emotional competencies such as emotional awareness, emotion regulation and social awareness.” Please find the citation basis for this sentence.

  1. Some expressions and arguments are not self-justified, and there is room for improvement in logical coherence and rigor.

5.1 “Drawing on knowledge from CLIL based literature and SEL we seek to develop tools that may be used by teachers and that lead their students towards autonomy of thought and a capacity for action, supported by positive self-perceptions while learning English.”This sentence is not closely related to the topic.

5.2 “It is also questionable, as Bur ton says, whether the content allows for more potential when communicating in a CLIL class compared to a foreign language class.” This sentence in this paragraph does not serve the previous argument.

5.3 “As students are at early stages of learning English, the language scaffolding plays an important role. The implementation of the sequence can be performed by any school teachers paying attention to the ways the content input activities are made accessible to students or English language teachers who add content to their language related exercises.” (Readers can't grasp the meaning of the sentence clearly, please use short sentences and rewrite the sentences to make it clearer)

Author Response

Response to the second review: Thank you very much for the constructive and precise feedback. Please find hereafter the responses and changes according to each comment.

  • Remark 1: We changed the title which can now read as follows:

Exploration of Teaching Lessons to Develop Students’ Emotional Competencies in English Classes: Are Emotions and Social and Emotional Learning Well-Suited to Content and Language Integrated Learning (CLIL) Lessons?

  • Remark 2: We thank the reviewer for this important remark. We now added the following explanations in the second section:

As far as the school’s classroom teaching is concerned, regardless of any courses or subjects, teachers can carry out teaching work around three objectives: first, developing skills and knowledge; second, working on emotions, attitudes and values; and third, making the students conscious of the learning process and strategies. The emotions intervene in the process of helping them achieve these goals and the integrated learning is part of the teaching.

They can help achieve the learning objectives in an English class, such as productive (speaking/writing) or receptive skills (listening/reading), acquiring the knowledge about the language (e.g., spelling, grammar), the meaning-making strategies to understand, speak or become aware of the learning itself (learning seen as a process or a result).

  • Remark 3: We are grateful to the reviewer for pointing out this relevant aspect. We now added further considerations regarding the development of social and emotional competencies. These precisions can now be read as follows at the beginning of section 6.

Research has shown that the identification of mixed emotions (e.g., I am both anxious and happy to give this talk about my favorite sport) progresses between the ages of 7 and 10 (e.g., [34]). However, other research has highlighted that 15-year-olds still have a fair amount of difficulty recognizing fear and sadness (e.g., [35]). It could therefore be particularly interesting to enrich the emotional lexicon, even more so from the age of 10-11 years and in a language context. For example, students have the opportunity to learn vocabulary from the new language while learning to better identify their own emotions. This work on the granularity of emotions, for example in terms of their valence (from unpleasant to pleasant) and their intensity (weak to strong), can be done via a grid (mood meter) allowing the teacher and the children to identify the mood they are feeling [36,37]. These elements may be considered as basic pillars to assist in the regulation of emotions which can be difficult, especially during adolescence (e.g., [39]).

Added references:

Pons, F.; Harris, P.L.; de Rosnay, M. Emotion Comprehension between 3 and 11 Years: Developmental Periods and Hierarchical Organization. Eur. J. Dev. Psychol. 2004, 1, 127–152, doi:10.1080/17405620344000022.

Theurel, A.; Witt, A.; Malsert, J.; Lejeune, F.; Fiorentini, C.; Barisnikov, K.; Gentaz, E. The Integration of Visual Context Information in Facial Emotion Recognition in 5- to 15-Year-Olds. J. Exp. Child Psychol. 2016, 150, 252–271, doi:10.1016/j.jecp.2016.06.004.

Hoffmann, J.D.; Brackett, M.A.; Bailey, C.S.; Willner, C.J. Teaching Emotion Regulation in Schools: Translating Research into Practice with the RULER Approach to Social and Emotional Learning. Emotion 2020, 20, 105–109, doi:10.1037/emo0000649.

Richard, S.; Gay, P.; Gentaz, É. Pourquoi et comment soutenir le développement des compétences émotionnelles chez les élèves âgés de 4 à 7 ans et chez leur enseignant.e ? Apports des sciences cognitives. Raisons Educ. 2021, 25, 261–287.  

Aïte, A.; Cassotti, M.; Linzarini, A.; Osmont, A.; Houdé, O.; Borst, G. Adolescents’ Inhibitory Control: Keep It Cool or Lose Control. Dev. Sci. 2018, 21, e12491, doi:10.1111/desc.12491.

  • Remark 4.1: It appeared as useful to keep essential elements from Sander (2013, p.16) and Scherer’s work (2005, e.g., p.702, 705) as they are important references in the field. We now rewrote this phrase as follows:

Generally speaking, emotions are characterized by a strong intensity, a short duration, an occurrence due to a clearly identifiable stimulus; in contrast, moods are of weaker intensity, longer duration and sometimes appear without a specific trigger. Emotions generate different reactions, notably physiological and behavioural (see e.g., [1,2] for more details regarding the components of emotions and other affective phenomena).

  • Remark 4.2: We removed the term “live” in the sentence which can now read as follows:

…the way in which individuals gradually learn to live and manage pleasant and painful emotions has an impact on…

  • Remark 4.3: We added the following precisions and references:

Courses in the school curriculum that are directly aimed at developing these skills are however lacking in Switzerland’s curriculum [9] as well as in teachers' practices in France [10].

Added reference:

Santé publique France Les Compétences Psychosociales: Un Référentiel Pour Un Déploiement Auprès Des Enfants et Des Jeunes. Synthèse de l’état Des Connaissances Scientifiques et Théoriques Réalisé En 2021.; Santé publique France: Saint-Maurice, 2022; ISBN 979-10-289-0770-9.

  • Remark 4.4: We now explicitly added the reference at the end of the sentence.

  • Remark 5.1: We put this sentence at the beginning of the paragraph.

Drawing on knowledge from CLIL-based literature and SEL, we seek to develop tools that may be used by teachers and that lead their students towards autonomy of thought and a capacity for action, supported by positive self-perceptions while learning English.

  • Remark 5.2: We agree with this remark and have now removed this sentence.
  • Remark 5.3: We have now rewritten this part which can now read as follows:

The implementation of the sequence can be performed by any school teachers paying attention to the ways the content input activities are made accessible to students. It can also be used by English language teachers who add content to their language related exercises.

Reviewer 3 Report

The article proposes the Content and Language Integrated Learning (CLIL) method for Engish lessons for non-English pupils, focusing on the simultaneous learning of language and emotional competencies. This seems to be really a perfect or at least promising match since in language courses there is great room for picking up emotional competencies. 

Remarks:
- There is always an obvious trap: students want to say something and sometimes definitely not their real, own emotions. In some (and unfortunately, not rare) cases they may become targets of bullying after showing emotions. They will probably repeat the answers of the majority.

- Artistic works require very skilled students. It is near impossible for a 10 years old pupil to create something (not a simple draw) carrying emotions correctly. Moreover, the cost/benefit ratio is not appropriate: most of the time is taken by the technical part, not devoted to expressing emotions.

-The last sentence of the Conclusions section is false or at least too general: teachers of STEM subjects are also working on both the curriculum and the soft skills of the students, but this method is probably not suitable for them.

Abbreviations: In my opinion, abbreviations should be avoided in the title (CLIL) and in the abstract, but 'CLIL' is explained only on page 2. In the Conclusions, the abbreviation 'RULER' is not explained.

Technical issues:
- Page 6: The site requires a study code: http://www.unige.ch/cisa/properemo/minipons/example.php
- Page 8: Figure 1 is of low resolution, the numbers in the bottom right corners are misleading.

Author Response

Response to third review: Thank you very much for the constructive and positive remarks and suggestions. Please find hereafter responses and changes according to each comment.

  • Following the first remark, we added the following sentence at the beginning of subsection 6.3. in a footnote:

Asking students to express their emotions may cause them to say something other than their real emotions or conform to the majority's responses. In some cases, showing certain emotions could lead to bullying. A crucial point may be to create a supportive classroom environment by setting clear rules (and enforcing them) in particular exercises. Non-Violent Communication and the expression of positive emotions of gratitude are easier to implement and work towards the installation of harmonious social relationships and an optimal classroom climate (e.g., [33,34]).

  • Following the second remark, we now added the following sentence:

Creating artistic works that properly convey emotions can be difficult. Teachers should thus focus on discussions that can follow such work: for example, they can ask "what makes you say that this work is related to these emotions? How would you have done it yourself to express these emotions?".

Following the third remark, we now added this precision in the last sentence:

If teachers of foreign languages want to work on both curriculum and the development of such overarching skills, the CLIL approach appears to be a promising match.

  • Abreviations are now explained; CLIL directly in the title and RULER in the text as follows:

Several approaches such as RULER (an acronym for Recognizing, Understanding, Labeling, Expressing and Regulating emotions) exist to help teachers and educators develop their own emotional competencies and teach emotional competencies to children from kindergarten through high school [10].

  • Technical issues: We changed Figure 1 and the link accordingly.

Round 2

Reviewer 3 Report

The authors addressed well all the issues pointed out in the 1st round.

Author Response

Thank you again for your remarks and suggestions!